# Effects of initial microbial biomass abundance on respiration during pine litter decomposition

**Michaeline B. N. Albright**[1]*, **Andreas Runde**[1], **Deanna Lopez**[1], **Jason Gans**[1],
**Sanna Sevanto**[2], **Dominic Woolf**[3], **John Dunbar**[1]

**1** Bioscience Division, Los Alamos National Laboratory, Los Alamos, NM, United States of America, **2** Earth and Environmental Sciences Division, Los Alamos National Laboratory, Los Alamos, NM, United States of America, **3** College of Agriculture and Life Sciences, Cornell University, Ithaca, NY, United States of America

* malbright@lanl.gov

**Data Availability Statement:** Unprocessed sequences are available through NCBI's Sequence Read Archive (PRJNA601499)

## Abstract

Microbial biomass is increasingly used to predict respiration in soil organic carbon (SOC) models. Its increased use combined with the difficulty of accurately measuring this variable points a need to directly assess the importance of microbial biomass abundance for carbon (C) cycling. To test the hypothesis that the initial microbial biomass abundance (i.e. biomass abundance on new plant litter) is a strong driver of plant litter C cycling, we manipulated biomass abundance by 10 and 100-fold dilution and composition using 12 source communities on sterile pine litter and measured respiration in microcosms for 30 days. In the first two days of microbial growth on fresh litter, a 100-fold difference in initial biomass abundance caused an average difference in respiration of nearly 300%, but the effect rapidly declined to less than 30% in 10 days and to 14% in 30 days. Parallel simulations with a soil carbon model, SOMIC 1.0, also predicted a 14% difference over 30 days, consistent with the experimental results. Model simulations predicted convergence of cumulative $CO_2$ to within 10% in three months and within 4% in three years. Rapid microbial growth, evidenced by appearance of visible microbial mats on the litter during the first week of incubation, likely attenuates the effects of large initial differences in biomass abundance. In contrast, the persistence of source community as an explanatory factor in driving differences in respiration across microcosms supports the importance of microbial composition in C cycling. Overall, the results suggest that the initial abundance of microbial biomass on litter is a weak driver of C flux from litter decomposition over long timescales (months to years) when litter communities have equal nutrient availability. By extension, slight variation in the timing of microbial dispersal to fresh litter is likely to be a minor factor in long-term C flux.

## Importance

Microbial biomass is one of the most common microbial parameters used in land carbon (C) cycle models, however, it is notoriously difficult to measure accurately. To understand the consequences of mismeasurement, as well as the broader importance of microbial biomass abundance as a direct driver of ecological phenomena, greater quantitative understanding of the role of microbial biomass abundance in environmental processes is needed. Using

**Funding:** This work was supported by the U.S. Department of Energy, Office of Science, Biological and Environmental Research Division, under award number F260LANL2018 to JD. The funders had no role in study design, data collection and analysis, decision to publish, or preparation of the manuscript.

**Competing interests:** The authors have declared that no competing interests exist.

microcosms, we manipulated the initial biomass of numerous microbial communities across a 100-fold range and measured effects on $CO_2$ production during plant litter decomposition. We found that the effects of initial biomass abundance on $CO_2$ production was largely attenuated within a week, while the effects of community type remained significant over the course of the experiment. Overall, our results suggest that initial microbial biomass abundance in litter decomposition within an ecosystem is a weak driver of long-term C cycling dynamics.

## Introduction

Microbial decomposition of plant litter is a key process in terrestrial carbon (C) cycling [1]. Although the dynamics of plant litter decomposition have been studied for decades [2], accurate prediction of ecosystem $CO_2$ fluxes remains a challenge because the controls on decomposition and their response to climate change are not fully understood [3]. Whereas early SOC models focused mostly on abiotic controls (e.g. substrate, moisture, and temperature) [4], an emerging body of research suggests that microbial factors play a key role in regulating decomposition [5–7].

General microbial abundance in soil is a common microbial property incorporated in SOC models as a factor affecting the rate of decomposition of various organic carbon pools [8, 9]. In models, biomass abundance is typically not measured but, rather, computed as a state variable whose dynamic size is determined by an interaction between the model's structure and parameters, and environmental conditions such as climate, soil physical and chemical properties, and organic matter additions [3, 8]. Microbial biomass is estimated in this fashion in part because of the cost and logistical barriers to measuring microbial biomass in soils globally. Field studies that have measured soil microbial biomass abundance show that it changes over space and time [10], and in response to changing climate [11–13] or disturbance [14]. Measurements of microbial biomass in soils across the globe range from fractions of a gram to 250 grams of biomass C per $m^2$ [15]. Biomass abundance is thought to be largely controlled by organic substrate and moisture availability [11, 16, 17].

Although microbial biomass is an established factor in SOC models, its relative importance as a driver of variation in soil C cycling is an ongoing question, especially within a single ecosystem. Some studies suggest microbial biomass abundance drives a large portion of variation in soil respiration [18, 19] and exerts a major influence over other ecosystem components by controlling energy and nutrient flow [20, 21]. In contrast, other studies posit only a small role of microbial biomass abundance in driving respiration, or propose complex relationships between biomass and abiotic factors [22, 23]. The importance of microbial biomass is generally inferred from correlative data [18, 19, 22, 23] where the contribution of co-varying abiotic soil characteristics and other biotic factors (e.g., microbial composition) is difficult to disentangle. Moreover, quantifying the impact of microbial biomass is difficult because biomass abundance is notoriously difficult to measure accurately [21, 24–26], measurements can vary 10-fold if community composition differs, and biomass is highly dynamic in some contexts. In dynamic contexts, the distinction between *intial* and *equilibrium* biomass abundance creates additional complexity. *Initial* abundance reflects the microbial state at the beginning of a dynamic process or at any time point where substantial changes in abundance are expected to follow, such as the addition of a new labile substrate. *Equilibrium* microbial biomass reflects the biomass level achieved after colonization of fresh substrate has occurred and the system has reached

what could be viewed as a carrying capacity or a steady state of biomass level. Uncertainties about the relative importance of initial biomass abundance emphasize the need for functional studies that attempt to manipulate microbial biomass independent of other abiotic and biotic variables and measure functional consequences.

We used microcosm experiments and modeling to assess the role of *initial* microbial biomass abundance on $CO_2$ dynamics during the early phase of plant litter decomposition. The early phase of litter decomposition is highly dynamic with rapid waves of microbial succession [27–29]. We examined this phase using pine litter decomposition in microcosms over a 30-day period. Although controlled microcosm studies lack the full realism of field studies, microcosms are ideal systems to explore general phenomena that are difficult to disentangle in more complex systems [6].

For the microcosm study, we extracted microbial communities from 12 geographically disparate soils and created 3 dilutions ($10^{-1}$, $10^{-2}$, $10^{-3}$) of each source community to obtain known relative differences in the initial quantity of biomass. Selecting a range of community types provides a more robust test of the dependence of litter decomposition dynamics on the initial abundance of microbial biomass. The 100-fold range of initial biomass abundance for each source community is similar to the range of variation estimated among global soils [15]. We then inoculated sterile pine litter in replicate microcosms with the three dilutions of each microbial community and tracked $CO_2$ flux over 30 days. Since biomass dilution can potentially also alter community diversity and composition, as well as successional dynamics, we conducted a second experiment to assess variation in microbial community composition among dilutions. We hypothesized that initial differences in microbial biomass abundance would cause substantial variation in respiration over a 30-day decomposition process, with higher biomass leading to significantly higher respiration.

## Materials and methods

### Experiment 1

**Biomass manipulation from 12 source communities.** Microbial inocula for litter were obtained from 12 surface soils (i.e., 12 source communities) by suspending each soil in buffer (S1 Table). Surface soils were collected during a larger sample collection across the southwest U.S. Samples were collected on public roadways at least 15 meters from the raod. The field sampling did not involve endangered or protected species. Soil geochemistry was not characterized, since soils were used to extract microbial communities to inoculate environmentally similar microcosms, while minimizing changes in geochemistry in the microcosms due to soil characteristics. We selected the 12 soils to represent a range of community types that can collectively provide a more robust test of the dependence of litter decomposition dynamics on the initial abundance of microbial biomass. To quantify the effect of initial biomass differences on $CO_2$ production, we made 3 serial dilutions ($10^{-1}$, $10^{-2}$, and $10^{-3}$) of each soil to provide known differences in the initial abundance of microbial biomass for each soil. The dilutions were created by suspending one gram of soil in 9 ml of phosphate-buffered saline (PBS) amended with $NH_4NO_3$ at 1 mg/ml, creating a 10-fold dilution. The 10-fold dilution was centrifuged for 1 minutes at max speed (16,500x g) and supernatant was decanted. The pellet was then resuspended in the same volume of $NH_4NO_3$ amended PBS buffer. These additional centrifugation and resuspension steps were performed in an effort to remove the bulk of soil chemistry effects. We then performed serial dilutions in PBS with $NH_4NO_3$ (1 mg/mL final concentration) to obtain 100- and 1000-fold dilutions of each source community.

**Initial biomass estimation among 12 source communities.** Our dilution approach allowed us to accurately determine relative variation in biomass *within* a source community.

To estimate biomass variation *among* source communities at the beginning of the experiment, we used both DNA quantification and bacterial and fungal plate counts. DNA extractions were performed with a PowerSoil 96-well plate DNA extraction kit (Mobio, San Diego, CA, USA). The standard protocol was used with two exceptions: 1) 0.3 grams of material was used per extraction, and 2) bead beating was conducted using a Spex Certiprep 2000 Geno-Grinder for three minutes at 1900 strokes/minute. DNA samples were quantified with an Invitrogen Quant-iT$^{TM}$ ds DNA Assay Kit on a BioTek Synergy HI Hybrid Reader. DNA quantities were used as a proxy for total microbial biomass.

Fungal and bacterial abundance was estimated from initial soil samples using plate counts. Serial dilutions of each soil were prepared in PBS, and 100-µl aliquots of appropriate dilutions were spread on 1/10 Trypticase Soy Agar (TSA) plates for bacterial counts or 1/10 TSA plates with chloramphenicol (100 ug/L) and gentamicin (50 ug/L) for fungal counts. Colonies were counted after incubating plates at 25˚C for 7 days.

**Microcosm construction and $CO_2$ sampling.** A total of 72 microcosms were constructed using 125 mL serum bottles containing *c.a.* 5 g of sand and 0.1 g of homogenized sterilized (autoclaved 1hr, twice) pine litter (*Pinus ponderosa*) finely ground with a Wiley mill (Thomas Scientific, Swedesboro NJ, USA. The serum bottles with pine litter were then autoclaved once for 1hr, to sterilize the microcosms. Microcosms were inoculated with 1.3 mL of microbial inoculum, with two replicate microcosms for each of the 36 inocula (12 source communities x 3 dilutions). Serum bottles were sealed with crimp caps and incubated in the dark at 25˚C.

After inoculation, $CO_2$ production in each microcosm was measured at 2, 5, 9, 16, 23, and 30 days by gas chromatography using an 490 Micro GC (Agilent Technologies, Santa Clara, CA, USA. Immediately after $CO_2$ measurements, the air in each microcosm was evacuated with a vacuum pump and replaced with ambient sterile-filtered air.

## Experiment 2

**Microcosm construction and $CO_2$ sampling.** While, the intent of the overall study was to measure the effects of initial biomass abundance on respiration, we set up an additional experiment to address concerns about the potential impacts of dilutions on microbial composition. The microcosm set-up and sampling protocols were identical to Experiment 1, except that a two-week pre-incubation phase was included [30] to reduce potential effects arising from pre-existing differences in the physiological state of microbial communities from the two soils (S010 and S018) tested in this experiment. Microbial inocula were added to microcosms that initially contained only 0.02 grams of litter and 5 g of sand. After 14 days incubation at 25˚C, an additional 0.1 grams of sterilized litter was added. During the priming phase $CO_2$ was measured on days 3, 7, and 14. After the 44-day (total) incubation, microcosms were destructively sampled to assess community composition. Microcosm samples were stored at -70˚C prior to DNA extraction (performed as in Experiment 1).

**Bacterial and fungal community taxonomic profiling.** Taxonomic profiling was performed by sequencing bacterial 16S rRNA and fungal 28S rRNA genes. The V3-V4 region of bacterial (and archaeal) 16S rRNA genes was amplified using primers 515f-R806 [31] and the D2 hypervariable region of fungal 28S rRNA gene was amplified using primers LR22R [32] and LR3 [33]. PCR amplifications for bacteria and fungi were performed using the same two-step approach [34]. In the first PCR, sample barcoding was performed with forward and reverse primers each containing a 6-bp barcode. Barcodes (6-mers) were designed in 2014 with standard rules: homopolymer length limited to 3nt, less than 4bp complementation between any pair of barcodes and no substantial complementation between the barcodes and the Illumina adaptors. 22 cycles with an annealing temperature of 60˚C were performed [35].

The second PCR added Illumina adaptors over 10 cycles with an annealing temperature of 65˚C. Amplicon clean-up was performed with a Mobio UltraClean PCR clean-up kit, following manufacturer's instructions with the following modifications: binding buffer amount was reduced from 5X to 3X sample volume, and final elutions were performed with 50 μl Elution Buffer. Following clean-up, samples were quantified using the same procedure as described in *Experiment 1* and pooled at a concentration of 10 ng per sample. A final clean-up step was performed on pooled samples using the Mobio UltraClean PCR clean-up kit. Samples were sequenced on an Illumina MiSeq platform with PE250 chemistry at Los Alamos National Laboratory. Unprocessed sequences are available through NCBI's Sequence Read Archive PRJNA601499.

Following sequencing, bacterial and fungal sequences were merged with PEAR v 9.6 [36], quality filtered to remove sequences with 1% or more low-quality (q20) bases, and demultiplexed using QIIME [37] allowing no mismatches to the barcode or primer sequence. Further processing was performed using UPARSE [38]. First, sequences with an error rate greater than 0.5 were removed, remaining sequences were dereplicated, singletons were excluded from clustering, Operational taxonomic unit (OTU) clustering was performed at 97%, and putative chimeras were identified *de novo* using UCHIME. Bacterial and fungal OTUs were classified using the Ribosomal Database Project (RDP) classifier. OTUs with less than 80% confidence in taxonomic assignment at the bacterial Phylum level or less than 100% confidence at the fungal Domain level were removed from the dataset. The excluded OTUs accounted for 0.3% of the bacterial data and 1.6% of the fungal data. Following quality control, the number of bacterial sequences ranged from 11,512 to 49470 per sample, representing a total of 922 OTUs. The fungal sequences ranged from 13,493 to 73,568 per sample, representing a total of 227 OTUs. Bacterial and fungal surveys were rarefied to 11,512 and 13,493 for comparisons of community diversity and composition.

## Statistical analyses

Pearson's correlations were used to assess correspondence between initial biomass estimates (DNA quantification (qPCR), fungal plate counts, bacterial plate counts) and respiration. To quantify the effects of initial biomass on cumulative respiration, we calculated the difference in cumulative $CO_2$ produced in the highest compared to lowest dilution for each source community at each timepoint and then calculated the average across the 12 source communities. To test for differences across the treatments and estimate the variance explained by each treatment in driving variation in univariate metrics (i.e. $CO_2$ production, richness, Shannon diversity) we used a nested ANOVA design with inoculum type as the main fixed factor and initial biomass as a nested factor within inoculum. Post hoc Tukey HSD tests were conducted to assess significant differences in richness and Shannon diversity across sample groups. The effects of initial biomass ($10^{-1}$, $10^{-2}$, $10^{-3}$ dilutions) and microbial community composition (i.e., the source community) on $CO_2$ production were assessed during each measurement period (day 0–2, day 2–5, day 5–9, day 9–16, day 16–23, day 23–30). The ANOVA and post hoc Tukey analyses were conducted in the R software environment (v3.5.1) [39].To examine the effect of initial biomass on the temporal pattern of respiration, the average "daily" $CO_2$ (i.e., the $CO_2$ measured at a given timepoint, not the cumulative $CO_2$ from the beginning of the decomposition process) was calculated across the 12 source communities for each level of initial biomass.

To assess the contribution of treatments in driving variation in bacterial and fungal community composition, we performed a permutational multivariate analysis of variance (PERMANOVA)[40]. As with the univariate version, this multivariate model included source community type as the main fixed factor and initial biomass as a nested factor within

community type. We estimated the percent of variation that could be attributed to each significant term for both the PERMANOVA (as described in [41]) and ANOVA analyses [42] as a percent of the Mean Square. To quantify the relative variability in microbial composition within each dilution (i.e. $10^{-1}$, $10^{-2}$, and $10^{-3}$), we measured the average distance to the centroid within each group using a test for homogeneity of dispersion [40].

## SOC modeling

A microbial SOC model, SOMIC 1.0 [43], was used to predict respiration rates and microbial biomass as a function of time in treatments with varying initial microbial biomass. Because the litter was placed on the soil surface in the incubation experiment, the SOMIC model was run for a single soil layer comprising an organic horizon with 0% clay content. The *Pinus ponderosa* litter was partitioned into fast and slow turnover pools (referred to as SPM and IPM, respectively, in the SOMic model) using the metabolic to structural material ratio calculated according to the DAYCent equation in which the metabolic fraction is calculated as 0.85–0.013 L/N, where L/N is the lignin to nitrogen ratio [44]. We assumed a value of L/N of 45.7 [45]. Microbial biomass in the 1x dilution treatment was assumed to be 2.3% of the carbon present in the plant litter, based on the mean value for continuous monocultures in Anderson (1989) [46]. The litter was assumed to have 48.94% carbon content, based on the average value for pine needles in the Phyllis 2 database [47]. A constant moisture content of 50% was assumed in the litter during the incubation simulation.

## Results

### Experiment 1: Effect of initial biomass and microbial composition on $CO_2$ flux during litter decomposition

Source soil DNA concentrations varied 50-fold, ranging from 0.5 to 25.9 ng/μL (average = 11.07±2.7 ng/μL; Fig 1A). Bacterial and fungal counts also varied widely among the source soils (S1 Fig). Among the source communities cumulative $CO_2$ production ranged from 187.1± 8.5 mg/g litter to 260.5±7.6 mg/g litter (Fig 1B). Cumulative respiration was not significantly correlated with any measurement of initial biomass (DNA concentration, $R^2$ = -0.13, P = 0.68; fungal plate counts, $R^2$ = -0.05, P = 0.87; bacterial plate counts, $R^2$ = 0.04, P = 0.91). The 100-fold difference in initial biomass ($10^{-1}$ versus $10^{-3}$ dilutions), created an average difference in $CO_2$ of 289 ± 66% at day 2. The difference in cumulative $CO_2$ declined to 25 ± 21% by day 9. Over the cumulative 30 day period, the large initial variation in biomass led to an average drop in total $CO_2$ of only 14.1 ± 2.1%, ranging from 1.7 and 25.5% (Fig 1C, S2 Fig).

During the first two days of decomposition, respiration was strongly driven by the initial biomass abundance and by initial community composition (nested ANOVA; initial biomass [source community]: $F_{24,36}$ = 13.7, P<0.001; source community: $F_{11,36}$ = 30.3, P<0.001). In that time interval, the two treatment factors explained 92% of estimated variation in $CO_2$ production, with the initial biomass explaining 52% and the source community explaining 40% (Fig 2). The effect of initial biomass attenuated rapidly; by day 16 (measurement interval: days 9–16), initial biomass was not a significant driver of differences in $CO_2$ production across microcosms. This trend persisted in the final two weeks, where source community remained the only significant factor (nested ANOVA; initial biomass[source community]: $F_{24,36}$ = 0.5, P = 0.96; source community: $F_{11,36}$ = 6.4, P<0.001), accounting for ~50% of variation in $CO_2$ production (Fig 2).

Average daily respiration increased rapidly immediately following inoculation of pine litter, peaked, and then decreased over time (Fig 3). For communities with the highest initial biomass

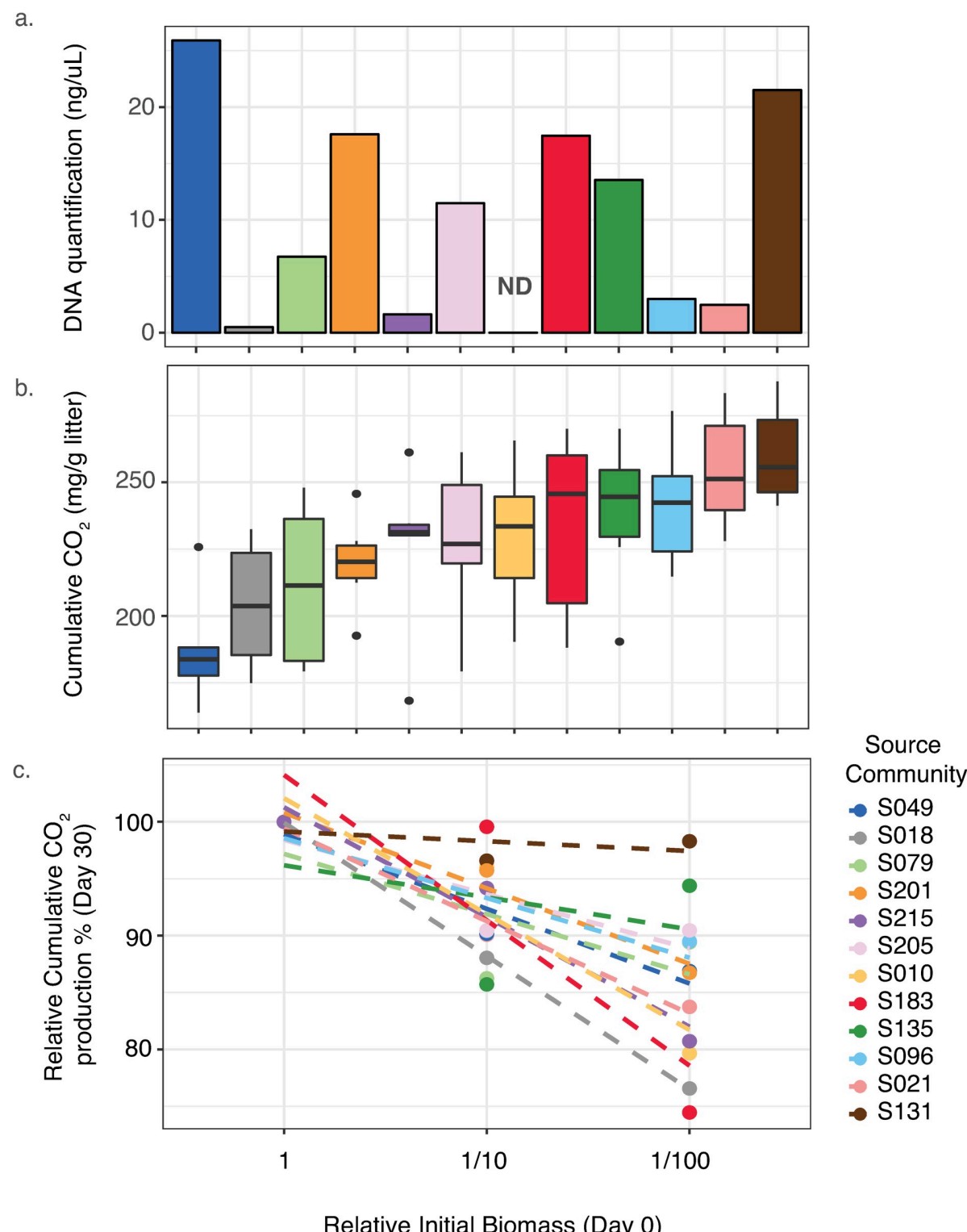

**Fig 1. Variation in biomass and respiration among source communities.** The source communities were extracted from 12 different soils. **a)** Biomass approximation from DNA concentration (ng/uL) from undiluted soil extractions. ND indicates no data. **b)** Cumulative respiration (mg $CO_2$/gram litter) after 30 days for each community type across all biomass dilutions. For each boxplot, the line in the box shows the median cumulative $CO_2$, with the endpoints showing the 25% and 75% quartile range. The whiskers show the 0% and 100% quartile range. Separate points outside of whiskers show outliers within a dilution. **c)** Relative cumulative respiration (by day 30) for each source community normalized by the cumulative $CO_2$ from the least diluted version ($10^{-1}$ dilution) of each source community.

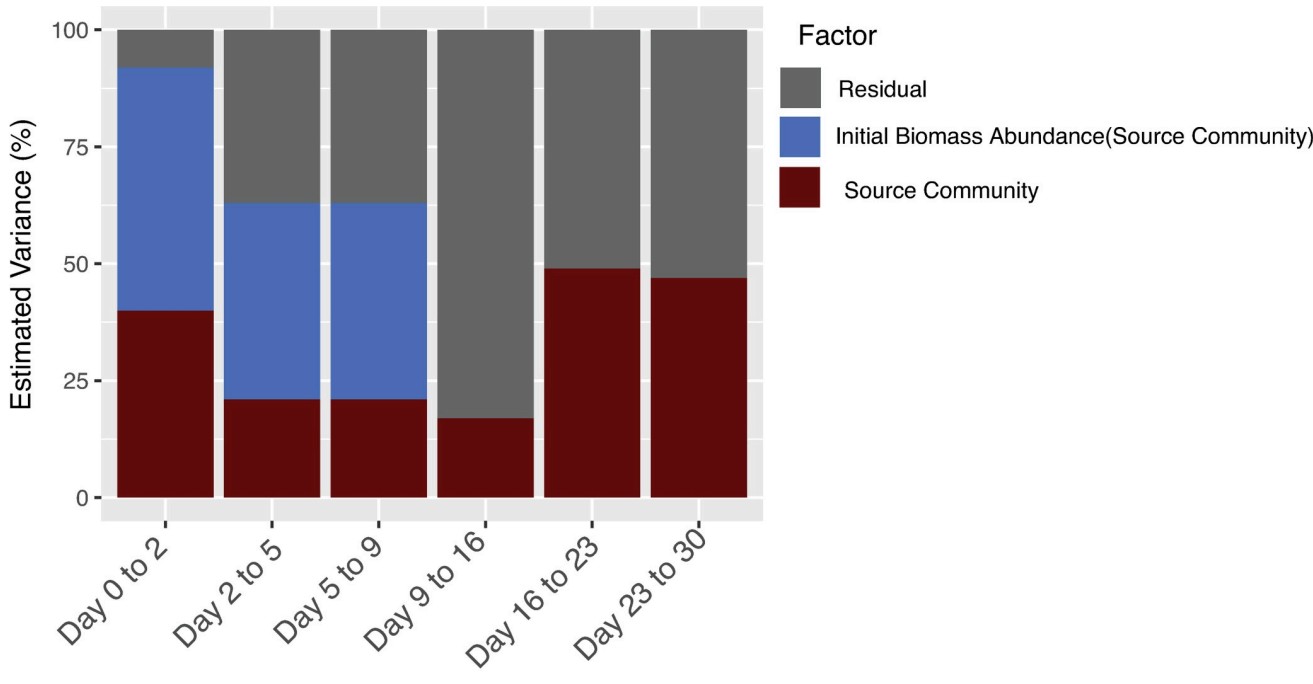

**Fig 2. Percentage of variation in respired $CO_2$ explained by initial biomass abundance (dilution level) and initial community composition (source community) for each time interval.**

level, the average daily respiration peaked by day 5, whereas respiration from communities with the lowest initial biomass level peaked by day 9 (Fig 3B).

**SOC modeling.** The SOMIC model predicted a 14.4% lower cumulative respiration after 30 days for communities that began with a 100-fold lower abundance of microbial biomass (Fig 4A). This is consistent with the experimentally observed 14.1 ± 2.1% lower cumulative $CO_2$ for the $10^{-3}$ treatment relative to $10^{-1}$. Extending the time scale, model simulations predicted convergence of cumulative $CO_2$ to within 10% at three months and 3% at three years. Modeled respiration rates peaked after 4.2 days and 9.5 days for communities with initial biomass levels of 1 and 1/100, respectively (Fig 4B). The model showed that microbial biomass in the two treatments gradually converged to a similar value over time. In the community with high initial biomass, microbial biomass increased as decomposition began, reaching a maximum after twelve days, whereas in the community with 1/100 initial biomass, microbial biomass continued to increase until near the end of the 30 days (Fig 4C). At 30 days the predicted microbial biomass in the two communities differed by 21%.

## Experiment 2: Dilution of initial biomass impacts microbial community diversity and composition

Although the second experiment included a two-week acclimation period to minimize effects from potential pre-existing differences in the physiological state of the soil communities, this did not alter respiration dynamics (S3 Fig). As expected, microbial community richness, diversity, and composition differed between by the two source communities (S2 Table). For both source communities, changes in relative initial biomass led to significantly different bacterial richness (nested ANOVA; $F_{4,12}$ = 6.8, p = 0.004) and fungal richness (nested ANOVA; $F_{4,12}$ =

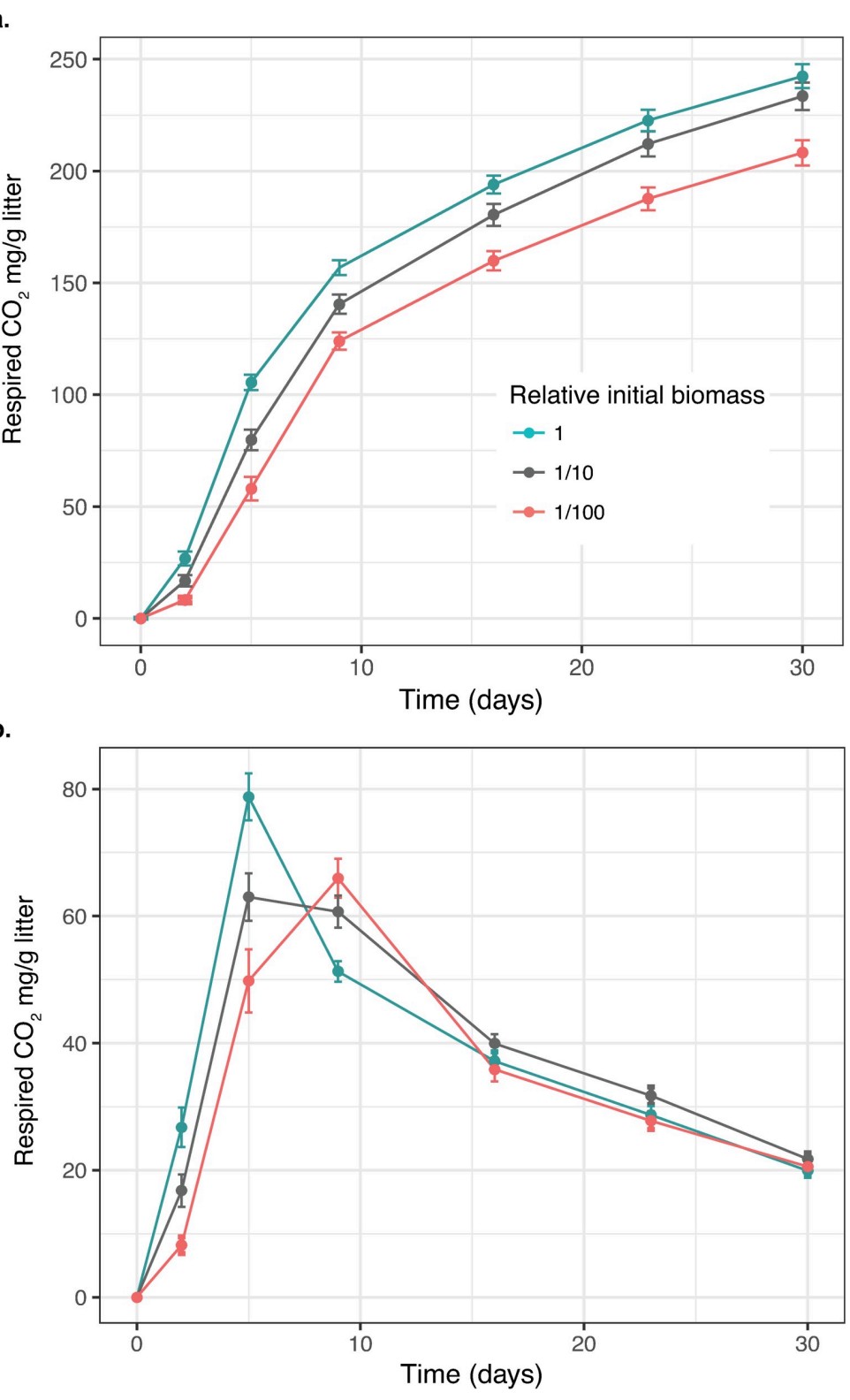

**Fig 3. Effect of initial biomass abundance on respiration. a)** Cumulative respiration across dilutions over time. **b)** "Daily" respiration, the respired $CO_2$ (mg/g litter) measured at a given timepoint, over time. Each point is the average respired $CO_2$ (mg/g litter) among 12 communities (2 replicates each) that were inoculated on litter at the indicated initial relative abundance level (1, 1/10, or 1/100). Error bars are 1 standard deviation (n = 12 per dilution).

**a** **Cumulative respiration**

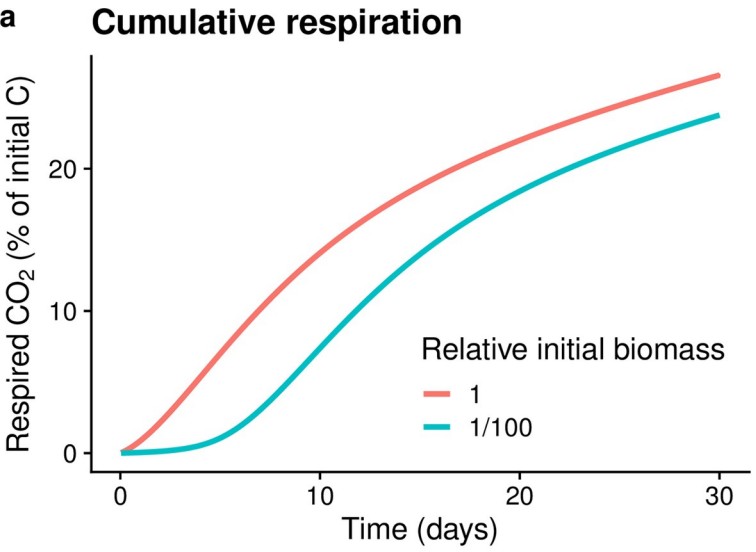

**b** **Daily respiration**

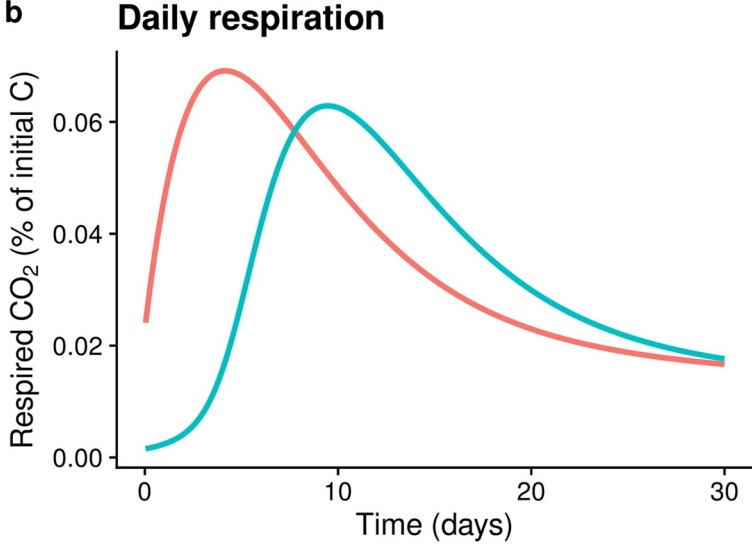

**c** **Microbial biomass**

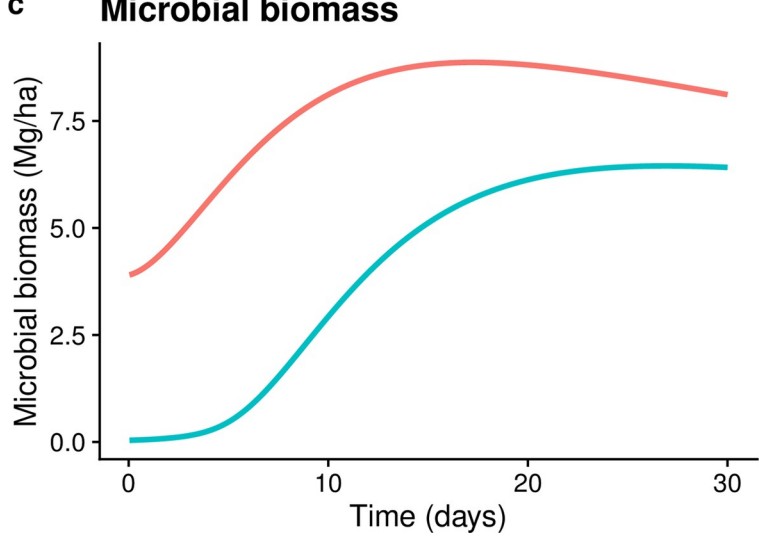

**Fig 4.** SOMIC 1.0 modeling of the effects of 100-fold differences in initial biomass abundance on **a)** Cumulative respired $CO_2$ over 30 days **b)** "Daily" respired $CO_2$ **c)** Temporal changes in microbial biomass.

7.7, p = 0.002) in the microcosms at day 30. The richness pattern showed a consistent decline in bacterial, but not fungal taxa by dilution (Fig 5A and 5B). For each source community, decreased initial biomass led to decreased bacterial diversity measured at the end of the 30-day litter incubation (nested ANOVA; $F_{4,12} = 50$, p<0.001) (Fig 5C), but did not alter fungal diversity (nested ANOVA; $F_{4,12} = 0.7$, p = 0.63) (Fig 5D). Furthermore, initial biomass significantly altered bacterial composition (nested PERMANOVA; $F_{2,14} = 6.3$, p = 0.001) (Fig 5E, S4A Fig), but not fungal community composition (nested PERMANOVA; $F_{2,14} = 1.7$, p = 0.07) (Fig 5F, S4B Fig). Variation in bacterial composition among samples was driven primarily by the source community type (PERMANOVA; estimated variation 51%), whereas the initial biomass (dilution treatment) had a smaller effect (PERMANOVA; estimated variation, 35%). Lastly, significantly greater variability in bacterial community composition occurred among replicates with the lowest

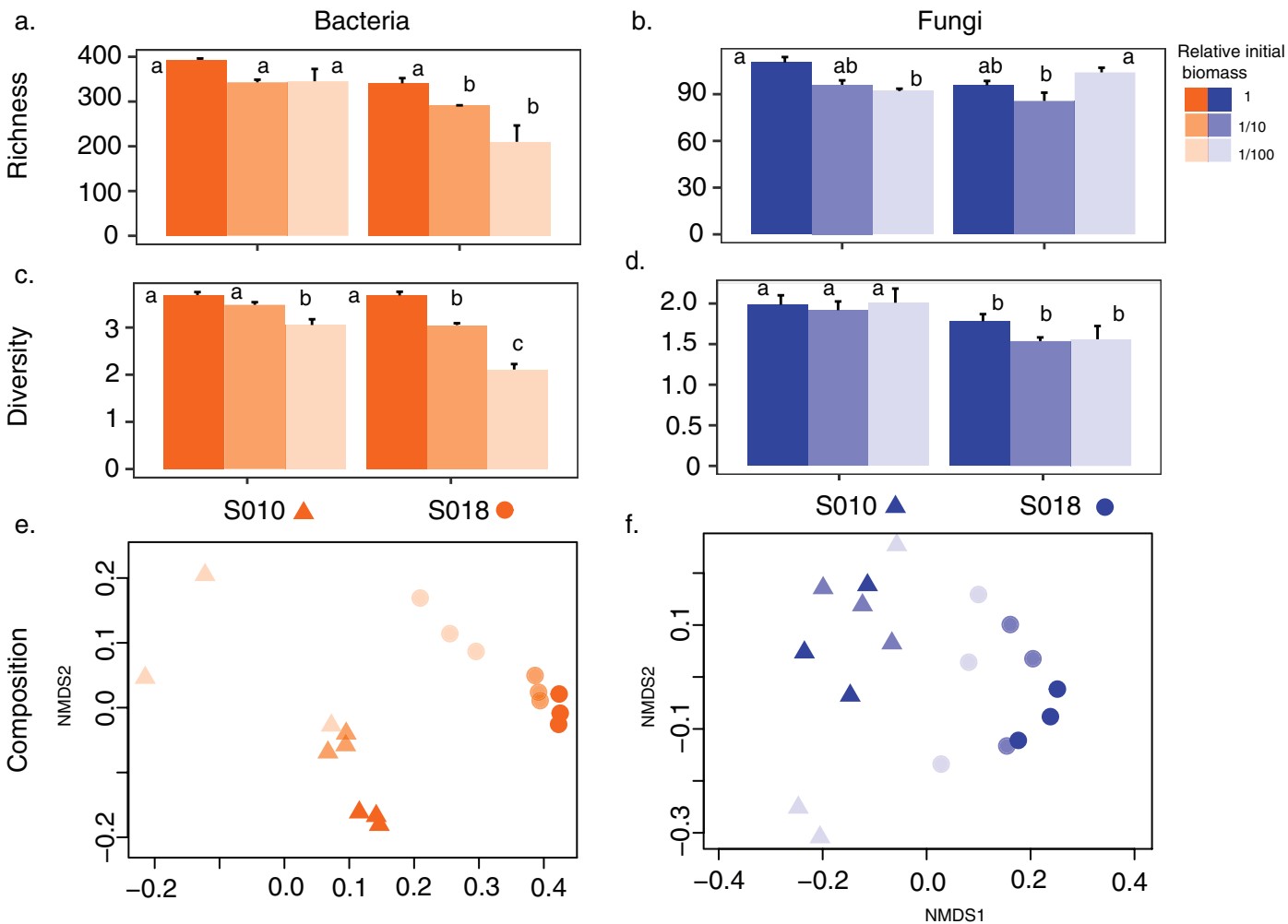

**Fig 5. Effect of initial biomass abundance of two source communities on the similarity of final microbial communities observed after 45 days of incubation in pine litter microcosms.** Two source communities (S010 and S018) known to differ in pine litter decomposition were tested. Bacteria and fungi in the final communities were assessed by profiling 16S and LSU rRNA gene sequences, respectively. Average OTU richness for **a)** bacteria and **b)** fungi. Average Shannon diversity for **c)** bacteria and **d)** fungi. Letters indicate significant differences across treatment groups from post hoc Tukey HSD tests. Compositional similarity of **e)** bacteria and **f)** fungi.

initial biomass compared to replicates with the highest initial biomass for both source communities, S018 and S010 (PERMDISP; p = 0.04 and p = 0.03, respectively) (Fig 5E, S4A Fig).

## Discussion

Within a single ecosystem, plant litter decomposition measured as cumulative $CO_2$ production or as litter mass loss can vary up to 2-fold (100%) after incubation periods ranging from 44 to 365 days [48, 49]. Identifying the drivers of such variation is important because it can improve modeling and management of soil carbon. Our combined use of experimentation and modeling to assess the importance of initial biomass abundance on pine litter provides quantitative information that has been absent in published literature. We found that 100-fold differences in the initial abundance of microbial biomass does create large variation in respiration (up to 767% in our study), but only over a short time period (several days). Over long timescales (weeks to years), the initial abundance of microbial biomass on litter is a weak driver of variation in cumulative respiration (Figs 2 and 4), as microbial biomass rapidly increases to the carrying capacity of the environment (Fig 4).

Attenuation of initial microbial biomass effects on litter decomposition was rapid in our microcosm study under constant conditions. In our study, the effect of initial biomass differences on cumulative respiration declined from an average 266% difference at day 2 to only 14% difference by day 30. Parallel modeling predicted similar results with only a 14% difference at day 30, declining to less than 4% over 3 years under constant environmental conditions. In addition, peak respiration rates occurred over a similar time frame in our experiment and model, indicating model dynamics operate at similar rates to the measured values in the microcosm experiments (Figs 3 and 4). The SOMIC model provided further insights including temporal dynamics of biomass and respiration over a much longer time scale. Our results support a previous soil microcosm experiment that used different methods to alter initial microbial biomass, but found soil organic matter, not initial microbial biomass, was the main driver of respiration differences over a 42-day incubation period [50]. Additionally, another experiment focused on later stages of decomposition, soil organic carbon (SOC) mineralization, found that mineralization proceeded at the same rate in soils where microbial biomass was decreased by greater than 90% using fumigation [51]. In nature, the rate of attenuation will likely depend on numerous factors including the magnitude and rate of microbial dispersal, litter colonization rates, microbial community composition (i.e., some types of communities may attenuate initial biomass effects faster than others), substrate conditions, and environmental conditions [52, 53]. A key question in nature is the magnitude of initial biomass differences expected within a single ecosystem. Whereas the 100-fold range tested in our study is relevant at the global scale [15], at a local scale the initial abundance of biomass on fresh litter may vary over a much smaller range (e.g., 2-fold; [52]).

Our use of microcosms enabled a test that cannot be performed effectively with natural litter in a field study—manipulating biomass abundance in a similar environment with 12 distinct communities and precise measurement of cumulative respiration. However, the microcosm approach involves tradeoffs. In this study the use of constant temperature and moisture likely accelerated decomposition, compared to fluctuating conditions in nature [54]. Nonetheless, results from a recent field study suggest that even in fluctuating environmental conditions initial biomass plays a minor role in driving decomposition dynamics [55]. In this field study a microbial community from desert plant litter with low initial inoculum biomass over 18-months carried out decomposition at a similar rate a microbial community from a grassland ecosystem with high initial inoculum biomass when inoculated onto the same plant litter substrate in microbial cages in the field [55]. Other tradeoffs expanded on below may be

worth considering although we do not expect them to alter the general conclusions about initial biomass abundance effects. For example, our use of pre-sterilized litter enabled manipulation of biomass abundance but eliminated the phyllosphere microflora, which contribute to decomposer communities in nature along with soil and rainwater microflora [24, 52]. We suspended source soil microbial communities into liquid and inoculated them onto plant litter in a single dispersal event analogous to heavy rain event that suspends soil microflora and deposits them on litter by splash or by surface flooding. The source communities were from surface soils that are a source of litter-colonizing organisms in dryland ecosystems in the western US. These ecosystems have sparse plant cover, where dispersal of microbes from exposed surface soil by rain and wind is a likely mechanism of plant litter colonization. Our microcosm litter communities had higher diversity (a few 100 taxa) than communities used in prior microcosm studies (10's of taxa) that yielded important insights into microbial processes, particularly decomposition [6, 56–58], but lower diversity compared to some previously studied natural plant litter systems (100's of taxa) [24, 59]. Despite these tradeoffs, the experimental results were consistent with results from a soil carbon model (SOMIC) that has been validated with natural field soils [43] and with findings from a field study where differences in initial biomass from source microbial inoculum had little impact on decomposition over a 1.5 year period [55]. This suggests the microcosm approach was suitable to test a fundamental biological process—the capacity for rapid microbial growth to attenuate the consequences of initially large differences in microbial biomass.

## Relative effects of biomass versus community composition on respiration

Separating the effects of microbial biomass and microbial composition is a challenge. Microbial biomass measurements are often influenced by microbial composition. Techniques used to measure microbial biomass abundance include phospholipid fatty acids (PFLAs) [60], substrate induced respiration (SIR) with selective inhibition of bacteria or fungi [61], DNA-based approaches such as qPCR [62, 63], growth-based measurements [64], and particle counts by flow cytometry [65, 66]. These techniques are affected by species' characteristics [24, 67]. For example, qPCR measurements of bacterial biomass can vary more than an order of magnitude owing to species-specific differences in the copy number of the qPCR gene targets, and a similar range of error has been demonstrated with other common measurement techniques [24]. Because of the measurement limitations, differences in decomposition dynamics cannot be attributed to biomass abundance (either initial or equilibrium abundance) unless microbial composition has been largely ruled out. For example, Bradford et al. (2017) estimated microbial biomass by SIR over a 24-hour incubation and concluded that microbial biomass variation accounted for the 2 to 5-fold variation in decomposition rates (% mass loss) of a homogenous plant litter substrate within 30 m transect field sites after a year-long incubation [48]. Our results show that up to 100-fold variation in microbial abundance at the time of sampling would be required to produce the observed SIR results in Bradford et al. [48]. We postulate that 100-fold variation in microbial biomass abundance is unlikely on litter at the same site after a year in the field and that variation in microbial community composition is likely a more important driver of decomposition rate. The scale of composition-dependent error in biomass measurements underscores the need to account for community composition in process dynamics.

To tease apart the effect of biomass abundance versus community composition on respiration dynamics, an ideal experiment would manipulate biomass and composition independently to control both factors. However, this can only be achieved with defined communities, which severely limits exploration of natural variation in soil microbial community composition. In our study, we attempted to manipulate the two factors biomass abundance via

dilutions and composition via use of of inoculum communities from 12 source. While our approach could not entirely disentangle biomass abundance and composition (see next section), the approach did enable estimation of the relative impact of each factor in a novel way. During the first two days of the experiment, 52% of the variation in respiration across microcosms was explained by the initial microbial biomass abundance, 40% was attributed to microbial composition, and only 8% was unexplained. By contrast, in the final few weeks of the experiment where microbial biomass had likely converged, composition still explained ~50% of the variation in respiration, while the remaining ~50% of variation was unexplained. The persistence of microbial composition as an explanatory factor supports its importance in driving respiration dynamics [5–7]. The increase in unexplained variation in respiration over time is likely due to stochastic phenomena that amplify community compositional differences causing divergence in function, in this case $CO_2$ flux, among replicates [68].

### Dilution impacts microbial community assembly

In our study, microbial community composition changed across dilutions, albeit to a smaller degree than composition varied among inocula (Fig 5E & 5F). Previous studies have tested the effects of dilutions on microbial communities and have found that dilutions can alter both diversity and function [69–72]. While we observed that increasing dilution of the initial community biomass resulted in reduced bacterial richness and diversity in the evolved decomposer communities (Fig 5), it is unlikely these small changes in composition significantly impacted respiration, as the dilution treatment had no significant impact on respiration rates in the final two weeks of the experiment when compositional divergence was likely greatest. Most dilution studies attempting to manipulate microbial richness use much higher dilutions, generally $10^{-6}$ to $10^{-9}$ [71]. In our study, the lowest initial biomass, $10^{-3}$ dilution, had significantly greater variability in bacterial community composition compared to the $10^{-1}$ dilution in both microbial communities where composition was measured (Fig 5). This was consistent with observations from previous microbial dilution studies that also show greater variability in composition from higher dilutions [71]. The result may arise either from stochastic founder effects fostered by the low initial biomass [71, 73] or by a greater opportunity for divergence through outgrowth, as the low biomass communities can undergo several more doublings before encountering substrate limitation.

## Conclusions

Our experimental and model simulation results suggest that initial microbial biomass abundance on litter is a weak driver of variation in respiration at the long time scales over which decomposition occurs. Community composition is likely a stronger long-term driver. Our findings emphasize the need to discover the microbial traits encapsulated in community composition that drive important variation in C cycling within and among ecosystems.

## Supporting information

**S1 Fig.** Estimated biomass for a) bacterial and b) fungi in source community soils using plate counts.
(DOCX)

**S2 Fig. Cumulative $CO_2$ production for all source communtiies after 30 days by relative intial biomass abundance.** For each boxplot, the line in the box shows the median cumulative $CO_2$, with the endpoints showing the 25% and 75% quartile range. The whiskers show the 0%

and 100% quartile range. Separate points outside of whiskers show outliers within a dilution.
(DOCX)

**S3 Fig. a)** Cumulative $CO_2$ (+- s.e.) production per day for source community soil 10 and soil 18 in Experiment 2. Initial biomass abundance shown in the same color. Acclimation period is shown with the red bar. Microbial communities were inoculated on 0.2 grams of pine litter, after 2 weeks (priming period), 1.0 grams of litter (5x initial amount) was added. **b)** Comparison of relative cumulative $CO_2$ (day 30) for each source community based on relative initial biomass (day 0) that was obtained through dilutions.
(DOCX)

**S4 Fig.** Relative abundance of abundant bacterial taxa at the order level, **b)** Relative abundance of abundant fungal taxa at the genera level. All individual samples are shown for each initial biomass abundance ($10^{-1}$, $10^{-2}$, $10^{-3}$).
(DOCX)

**S1 Table. Soils used to extract the twelve source microbial inoculum.**
(DOCX)

**S2 Table. Nested ANOVA for richness and diversity metrics and nested Permutational MANOVA for composition metrics.**
(DOCX)

## Acknowledgments

We thank David Myrold, Renee Johansen, and Cheryl Kuske for comments on previous versions of this manuscript.

## Author Contributions

**Conceptualization:** Sanna Sevanto, John Dunbar.

**Formal analysis:** Michaeline B. N. Albright, Andreas Runde.

**Funding acquisition:** John Dunbar.

**Investigation:** Michaeline B. N. Albright, Andreas Runde, Deanna Lopez.

**Methodology:** Jason Gans, Dominic Woolf.

**Project administration:** John Dunbar.

**Resources:** John Dunbar.

**Supervision:** Sanna Sevanto, John Dunbar.

**Visualization:** Michaeline B. N. Albright, Dominic Woolf.

**Writing – original draft:** Michaeline B. N. Albright.

**Writing – review & editing:** Michaeline B. N. Albright, Jason Gans, Sanna Sevanto, Dominic Woolf, John Dunbar.

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
