## [Decision Letter · Decision Letter 0]

7 Jan 2020

PONE-D-19-28820

Effects of initial microbial biomass abundance on respiration during pine litter decomposition

PLOS ONE

Dear Dr. Albright,

Thank you for submitting your manuscript to PLOS ONE. After careful consideration, we feel that it has merit but does not fully meet PLOS ONE’s publication criteria as it currently stands. Therefore, we invite you to submit a revised version of the manuscript that addresses the points raised during the review process.

We are sorry for the delay in the review process. Please have a look at the reviewers' useful comments.

We would appreciate receiving your revised manuscript by Feb 21 2020 11:59PM. To enhance the reproducibility of your results, we recommend that if applicable you deposit your laboratory protocols in protocols.io, where a protocol can be assigned its own identifier (DOI) such that it can be cited independently in the future. For instructions see: http://journals.plos.org/plosone/s/submission-guidelines#loc-laboratory-protocols

We look forward to receiving your revised manuscript.

Kind regards,

Riikka Rinnan, Ph.D.

Academic Editor

PLOS ONE

Reviewers' comments:

Reviewer's Responses to Questions

**Comments to the Author**

1. Is the manuscript technically sound, and do the data support the conclusions?

Reviewer #1: Yes

Reviewer #2: Yes

2. Has the statistical analysis been performed appropriately and rigorously? 

Reviewer #1: Yes

Reviewer #2: Yes

3. Have the authors made all data underlying the findings in their manuscript fully available?

Reviewer #1: No

Reviewer #2: No

4. Is the manuscript presented in an intelligible fashion and written in standard English?

Reviewer #1: Yes

Reviewer #2: Yes

5. Review Comments to the Author

Reviewer #1: This is a nicely designed, executed and described study showing how initial microbial biomass affects sterilized pine litter degradation in a microcosm experiment. I particularly like the combination of modelling and experimental approaches - it is very informative and shows that the SOMIC model works well also for litter.

The paper is very well written, all the parts are of adequate length and convey necessary information (with minor points in Materials and Methods). I don't have any concerns of methodological nature nor pertaining to data presentation.

I have only a couple of questions/suggestions of relatively minor importance:

1. The sequencing system could be better described: is it based on custom primers or native Illumina ones? Bioinformatics analysis description suggests the latter, but it is not clear.

2. Why have the Authors chosen Pearson's correlation? It assumes linear relation of variables, which might be true or not.

3. The Authors erroneously quote ANOVA instead PERMANOVA when stating that initial biomass affected bacterial community composition and did not affect fungal community (l. 315-316).

4. It is not stated how the percent of variance explained was calculated in PERMANOVA analysis. As % of MS or SS? Please specify in the M&Ms.

5. It is not stated in the text which soils were used in the Experiment 2 - it follows from figures, but please put this information in the text.

I would like to point out that the paper lacks the accession number for the SRA record containing reads generated in the study, but as the Authors state that the record is being prepared, I think it's fine.

Reviewer #2: In this manuscript, the authors use experimentation and modelling to assess the quantitative contribution of initial biomass as a driver of litter decomposition. These data are rare and as such the manuscript makes a good contribution to the field. It is generally a well written manuscript but the structuring could be improved to streamline the discussion of the results. I have some specific points below mostly surrounding improving the presentations of hypothesis and the results as well some questions on data analysis. Otherwise the data and the analysis in manuscript are sound and I endorse its publication in PLOS One after those concerns are met.

The authors highlight that the rationale of their hypothesis surrounding use of initial biomass abundance is the fact that microbial biomass is increasingly being used in soil C models. But as far as I understand, models often use biomass as a dynamic pool. So more than assessing the implications of biomass incorporation into models, I would say that the study is aimed at investigating the role of initial biomass in litter decomposition.

Order of results and figures is random. For eg. figure 3 shows respired CO2 over time and comes after cumulative CO2 was presented in figure 1 followed by analysis of variance. Figure 4 is presenting modelled data. I suggest structuring the results to present the actual data and patterns first followed by statistics and modelling outputs.

The authors do not show biomass shifts over the period of the incubation. Did the biomass change over time? They modelled it over time (Fig 4) and use those results to imply that biomass rapidly increases to the environment’s carrying capacity. I wonder if the predicted biomass numbers were verified by observed data. If it’s not available, that would be major limitation of the study.

It looks like the reduction in respiration with diluted initial biomass differed for the various communities used. Was the reduction higher in particular type of community?

Keywords do not accurately represent the manuscript

Fig 1a: What’s the rationale behind the use of bar charts to show DNA amounts. Also, there are no error bars shown.

Fig 3 and 4: The colour scheme in these figures is contradictory. It would also be ideal to merge these two figures to show 6 plots for the observed and modelled changes in respired CO2 over time.

Figure 5: Was this decrease in richness and diversity significant? It would be good for the reader to have that information in the figure.

6. PLOS authors have the option to publish the peer review history of their article (what does this mean?). If published, this will include your full peer review and any attached files.

Reviewer #1: No

Reviewer #2: No

---

## [Author Response · Author response to Decision Letter 0]

17 Jan 2020

Reviewers' comments:

Reviewer's Responses to Questions

Comments to the Author

3. Have the authors made all data underlying the findings in their manuscript fully available?

Reviewer #1: No

Reviewer #2: No

Data is now available in the NCBI SRA under Bioproject PRJNA601499

5. Review Comments to the Author

Reviewer #1: This is a nicely designed, executed and described study showing how initial microbial biomass affects sterilized pine litter degradation in a microcosm experiment. I particularly like the combination of modelling and experimental approaches - it is very informative and shows that the SOMIC model works well also for litter.

The paper is very well written, all the parts are of adequate length and convey necessary information (with minor points in Materials and Methods). I don't have any concerns of methodological nature nor pertaining to data presentation.

We thank the reviewer for their positive feedback.

I have only a couple of questions/suggestions of relatively minor importance:

1. The sequencing system could be better described: is it based on custom primers or native Illumina ones? Bioinformatics analysis description suggests the latter, but it is not clear.

We thank the reviewer for pointing out this missing information. We have added text regarding the custom primers in the methods (Lines 197-200).

2. Why have the Authors chosen Pearson's correlation? It assumes linear relation of variables, which might be true or not.

We wanted to test if initial estimates of biomass were directly correlated with respiration and we expected that if correlated, the correlation would be linear. We also visually inspected the graphs and we did not observe a non-linear trend or in fact any correlative trend. 

3. The Authors erroneously quote ANOVA instead PERMANOVA when stating that initial biomass affected bacterial community composition and did not affect fungal community (l. 315-316).

We than the reviewer for catching this error.

4. It is not stated how the percent of variance explained was calculated in PERMANOVA analysis. As % of MS or SS? Please specify in the M&Ms.

Cumulative respiration was not significantly correlated with any measurement

We have added this information in the M&Ms (Lines 250-251). Percent of variance was calculated as a % of MS.

5. It is not stated in the text which soils were used in the Experiment 2 - it follows from figures, but please put this information in the text.

We have added this information to the M&Ms text (Line 183).

I would like to point out that the paper lacks the accession number for the SRA record containing reads generated in the study, but as the Authors state that the record is being prepared, I think it's fine.

Data is now published in the NCBI SRA under Bioproject PRJNA601499

Reviewer #2: In this manuscript, the authors use experimentation and modelling to assess the quantitative contribution of initial biomass as a driver of litter decomposition. These data are rare and as such the manuscript makes a good contribution to the field. It is generally a well written manuscript but the structuring could be improved to streamline the discussion of the results. I have some specific points below mostly surrounding improving the presentations of hypothesis and the results as well some questions on data analysis. Otherwise the data and the analysis in manuscript are sound and I endorse its publication in PLOS One after those concerns are met.

We thank the reviewer for their positive assessment of the study.

The authors highlight that the rationale of their hypothesis surrounding use of initial biomass abundance is the fact that microbial biomass is increasingly being used in soil C models. But as far as I understand, models often use biomass as a dynamic pool. So more than assessing the implications of biomass incorporation into models, I would say that the study is aimed at investigating the role of initial biomass in litter decomposition.

We agree with the reviewer and we state these points in Lines 85-105 in the introduction. The important distinction that we make on Lines 97-105 is about the relative importance of initial versus equilibrium biomass. If both these biomass pools are equally important then experimentalists need to measure both in field studies, but if initial biomass abundance is only important over very short timescales then this is useful for experimentalists to understand to prioritize resource allocations. We have attempted to clarify further on line 103.

Order of results and figures is random. For eg. figure 3 shows respired CO2 over time and comes after cumulative CO2 was presented in figure 1 followed by analysis of variance. Figure 4 is presenting modelled data. I suggest structuring the results to present the actual data and patterns first followed by statistics and modelling outputs.

We agree with the reviewer that in many instances the reviewer’s proposed results structure is most logical, but we do not feel that this is the best approach for this paper. In this paper, the results/figures are not random, they are arranged by concept so that readers do not need to jump between key points. We also have 2 experiments which creates another logical split. The ordered concepts include: 1. Importance of variables tested (Initial biomass vs Source community) (Expt 1) 2. Observed vs Modeled Results (CO2 attenuation) (Expt 1) 3. Community composition (Expt 2). 

The authors do not show biomass shifts over the period of the incubation. Did the biomass change over time? They modelled it over time (Fig 4) and use those results to imply that biomass rapidly increases to the environment’s carrying capacity. 

We have added text (Line 42-43) to indicate that microbial growth was apparent in microcosms from visual inspection. This statement is also qualified in the text by a ‘likely’ thus we acknowledge that this is not measured. 

I wonder if the predicted biomass numbers were verified by observed data. If it’s not available, that would be major limitation of the study.

We are glad that the reviewer found the predicted biomass an interesting aspect of the paper. The key conclusions in the paper do not depend on the temporal biomass dynamics, as the focus of the study is on the impacts of initial biomass abundance. Modeling the biomass dynamics is a valuable conceptual addition, but not crucial quantitative data upon which the main conclusions were based.

It looks like the reduction in respiration with diluted initial biomass differed for the various communities used. Was the reduction higher in particular type of community?

The reviewer brings up an interesting point. Looking at this aspect of impact of community type would be interesting but this is beyond the scope of this study. One would need a means to group the 10 communities into 2 or 3 different types in order to do an analysis. We do not have a sensible way of defining community types with this data.

Keywords do not accurately represent the manuscript

We have altered key words to better reflect the content of the manuscript. (Lines 17-18)

Fig 1a: What’s the rationale behind the use of bar charts to show DNA amounts. Also, there are no error bars shown.

We chose to use bar charts because presenting DNA abundance in this way is common in published literature. There are no error bars because the data represent the original soil and multiple DNA extractions were not performed.

Fig 3 and 4: The colour scheme in these figures is contradictory. 

The only discrepancy that we can find is a red line that was dashed (Fig 3) and solid (Fig 4). We have revised Figure 3 to make the red line solid.

It would also be ideal to merge these two figures to show 6 plots for the observed and modelled changes in respired CO2 over time.

We thank the reviewer for this suggestion. We have added a panel so that Fig 3a and 3b correspond to Fig 4a and 4b. We have not combined the figure as we do not have experimental temporal biomass measurements (complimentary experimental data for Figure 4c). Figure 3a was originally in the supplementary material (Figure S3), so we have also revised the supplementary figure numbering.

Figure 5: Was this decrease in richness and diversity significant? It would be good for the reader to have that information in the figure.

We have added results from a posthoc Tukey HSD test to show significance in Figure 5.

---

## [Editor Report · Decision Letter 1]

27 Jan 2020

Effects of initial microbial biomass abundance on respiration during pine litter decomposition

PONE-D-19-28820R1

Dear Dr. Albright,

We are pleased to inform you that your manuscript has been judged scientifically suitable for publication and will be formally accepted for publication once it complies with all outstanding technical requirements.

With kind regards,

Riikka Rinnan, Ph.D.

Academic Editor

PLOS ONE

Additional Editor Comments (optional):

When supplying the final files, please make the following corrections:

Please mention post hoc tests that were made also in the description of statistical analyses. Also, describe the meaning of the letters (results from Tukey) in the figure legends.

L. 128. elast should be least
---

## [Editor Report · Acceptance letter]

30 Jan 2020

PONE-D-19-28820R1 

Effects of initial microbial biomass abundance on respiration during pine litter decomposition 

Dear Dr. Albright:

I am pleased to inform you that your manuscript has been deemed suitable for publication in PLOS ONE. Congratulations! Your manuscript is now with our production department. 

With kind regards,

on behalf of

Dr. Riikka Rinnan 

Academic Editor

PLOS ONE